# Prevalence of, Factors Associated with and Level of Dependence of Psychoactive Substance Use among Mekelle University Students, Ethiopia

**DOI:** 10.3390/ijerph17030847

**Published:** 2020-01-29

**Authors:** Azeb Gebresilassie Tesema, Znabu Hadush Kahsay, Gebrezgi Gidey Lemma, Welday Hagos Gebretsadik, Mamuye Mussie Weldemaryam, Gebrecherkos Gebregiorgis Alemayohu, Maree L Hackett

**Affiliations:** 1Health Education and Behavioral Science Unit, School of Public Health, Mekelle University, Mekelle 1871, Ethiopia; azeb18@gmail.com; 2Psychiatry Department, School of Medicine, Mekelle University, Mekelle 1871, Ethiopia; gebrezgi6@yahoo.com (G.G.L.); weldaypsycho@gmail.com (W.H.G.); mamuye6@yahoo.com (M.M.W.); 3Sociology Department, College of Business and Economics, Mekelle University, Mekelle 1871, Ethiopia; gebrecherkos.g@gmail.com; 4The George Institute for Global Health. Faculty of Medicine, University of New South Wales, Camperdown, New South Wales, M201, Sydney city 2052, Australia; mhackett@georgeinstitute.org.au

**Keywords:** substance use, alcohol, tobacco, khat, Mekelle University, college students, Ethiopia

## Abstract

Background: Psychoactive substance use is a major public health concern globally. Though youth attending higher education institutions are considered particularly vulnerable to psychoactive substances, there is a paucity of evidence in Ethiopia. We aimed to determine the prevalence of psychoactive substance use, factors associated with psychoactive substance use and level of dependence among Mekelle University undergraduate students in Ethiopia. Methods: An institution-based quantitative cross-sectional survey was used to randomly (using multistage sampling) invite 1220 undergraduate students in April and May 2017 to participate. Multinomial logistic regression was used to identify factors associated with psychoactive substance use. Level of dependence was determined using the WHO’s Alcohol, Smoking and Substance Involvement Screening Test classification. Results: Among the eligible students approached, 1214 participants completed the survey, which yields a 99.5% response rate. The lifetime prevalence of psychoactive substance use was 66.5% (95% Confidence Interval (CI) = 64% to 69%) while the current prevalence was 49% (95% CI = 46% to 52%). A history of, but not current, psychoactive substance use was reported by 18%, while 33.5% reported never having used psychoactive substances. The current prevalence of alcohol use was 35.5%, tobacco 7.8% and khat 5.7%. Of the current users, 17% (95% CI = 14% to 20%) were at a moderate to high risk of dependency. Being over 21 years of age (adjusted odds ratio (AOR) = 1.6, 95% CI = 1.37 to 2.25), male (AOR = 3.13; 95% CI = 2.26 to 4.34), living in urban areas (AOR = 2.39, 95% CI = 1.77 to 3.23), an Orthodox Christian (AOR = 7.55, 95% CI = 4.56 to 12.48), and being in their 3rd year (AOR = 2.3, 95% CI = 1.49 to 3.55), 4th year (AOR = 2.0, 95% CI = 1.2 to 3.51) and 5th year (AOR = 4.0, 95% CI = 2.81 to 7.67) at university were associated with currently using psychoactive substances. Being male and Orthodox Christian was associated with being an ex-smoker. Conclusions: Approximately half of Mekelle University undergraduate students were using psychoactive substances with almost one in five at risk of dependency. The likelihood of use increased with seniority. Evidence-based strategies are needed to prevent school-aged children from using psychoactive substances and university students becoming dependent on substances. Interventions designed to stop current psychoactive substance use may also have promise for reducing dependency.

## 1. Introduction

The World health organization (WHO) defines psychoactive substances as substances that, when taken in or administered into one’s system, affect mental processes. This includes alcohol, khat, tobacco and illicit drugs [1]. An estimated quarter of a billion adults globally use psychoactive substances at least once, of whom about 29.5 million experience disorders [2,3]. Literature indicates that psychoactive substance use attributes to negative social, physical and mental health consequences [4,5,6]. Those aged between 16 and 24 years are particularly vulnerable to psychoactive substance use [7,8]. Other associated factors include being at an age associated with experimentation, unemployed, having a low income, experiencing peer pressure, low self-esteem and living in a low or middle income country. About 80% of tobacco users are from low or middle income countries [9,10,11,12,13]. Despite a slight decline among the general population in the past decade globally [2] and nationally [11], recent evidence shows a higher prevalence of substance use among students attending higher educational institutions in Ethiopia and other low or middle income countries [12,13,14,15]. Following completion of pre-university education, students from areas of the country where substance use is common enroll at Ethiopian universities, including Mekelle University. Among Ethiopian men, 46% use khat, 27% alcohol and 4% tobacco [11]. In the Tigray region, where Mekelle University is based, 91% of the men and 71% of women consume alcohol [11].

Determining the prevalence of psychoactive substance use among youth underpins evidence for decision makers and program implementers to look at the trend of the practice and the dynamic webs of its negative consequences. Psychoactive substance use is concerning in students attending higher education institutions as it is associated with low educational achievement, conflict with parents, loss of friendship, financial hardship, limited creativity and intellect, risky sexual behavior, crime and violence, injuries and suicide [13,16,17]. Previous university-based studies indicated that students’ perceived performance on reading for longer hours, pleasure, socialization and relief from tension are some of the reasons for psychoactive substance use. Being male, more senior, coming from urban areas and having a family history of substance use were also associated with psychoactive substance use at university [18,19,20]. A recent population-based study among adolescents also shows that the perceived availability of drugs, a family history of use and peer pressure are positively correlated with substance use [21]. A previous qualitative study from Mekelle University also reveals that a range of individual, interpersonal (relational) and organizational factors drive university students toward psychoactive substance use [22]. However, there is a paucity of evidence regarding its prevalence and level of dependency in this vulnerable segment of the population in Ethiopia. Determining the prevalence, identifying the factors associated with psychoactive substance use and the level of dependence at university-level in Ethiopia may inform the design and implementation of cost-effective treatment and prevention programs. While offering help to substance users to quit is recognized as one of the six strategic interventions suggested by the World Health Organization (WHO) to reduce the burden of psychoactive substance use [21], there is limited evidence on the degree of dependency on the substance(s) used.

## 2. Methods

### 2.1. Study Aim, Design and Setting

We aimed to 1) determine the prevalence of substance use, 2) identify factors associated with substance use and 3) determine the level of dependence on substances used among Mekelle University students, in Northern Ethiopia. An institution-based quantitative cross-sectional study was conducted among regular undergraduate students in Mekelle University between 1 April and 30 May 2017. Mekelle University is located in Mekelle town, the capital of the Tigray regional state, Ethiopia. It has six campuses, with 26,747 undergraduate students [23]. Mekelle University established a substance users’ rehabilitation center in 2014.

### 2.2. Sample Size and Sampling Procedure

The sample size was determined using a single population proportion formula for cross-sectional surveys, based on a 22% prevalence of khat chewing from a study at Bahir Dar University [20]. Using a desired precision (d) of 0.03 with 95% confidence, a design effect of 1.5 and 7% inflation to compensate for the possible non-response rate, the final sample needed was 1220 respondents.

All regular undergraduate students attending at Mekelle University were eligible to participate. A multistage sampling technique was used to select the study participants. Out of eight colleges four colleges were randomly selected, and from each selected college, 50% of the departments were randomly chosen for a total of 15 departments. A sample frame for each department was prepared using the list of registered students in the selected departments, which was obtained from the University’s registrar office. Consequently, the required samples were proportionally allocated to the number of students in each selected department based on the list provided from the registrar’s office. Finally, computer-generated random sampling was employed to select the study participants.

### 2.3. Data Collection Process and Instrument

The questionnaire was prepared by reviewing the WHO students’ drug use survey questionnaire, the ASSIST guideline and other related literature [24]. The finally developed tool included three sub-sections: (1) socio-demographics, (2) self-reported substance use status and (3) level of dependence on the substance(s) they use. Substance use status and level of dependency on psychoactive substance use were measured using seven questions adopted from the “Alcohol, Smoking, and Substance Involvement Screening Test (ASSIST)” guideline, which specifically asked about
(1) all substances a participant had ever used in his/her life time; (2) frequency of use in the past three months preceding the survey; (3) frequency of experiencing a strong desire to use a substance(s) in the past three months preceding the survey; (4) frequency of health, social, legal or economic problems related to substance use in the past three months preceding the survey; (5) frequency of substance use interfering with students’ roles and responsibilities in the past three months preceding the survey; (6) if she/he has a friend or a relative or anyone else ever expressed concern about your use of the substance(s); and (7) if she/he has ever tried and failed to control, cut down or stop using the substance(s) [24]. The questions are in increasing order of specificity. The questionnaire explicitly asked questions about alcohol, cigarette, khat and cannabis use as these are recognized as the most frequently tried from other studies. This measurement enabled us to determine the lifetime and current prevalence of psychoactive substance use as well as the level of dependence on the substances under use. Participants indicated their response on the questionnaire using the pen and paper approach. Experts in health education and behavioral science, psychiatrists, a clinical psychologist and a sociologist developed the self-administered questionnaire in English. One language expert translated it into Amharic (the Ethiopian national official language); another language expert back-translated it to English to ensure consistency of terms and meaning. The final tool took 15–20 min to complete. The data collectors were trained for two days, and they approached the randomly selected students in a class arranged for the survey. Data collectors were in class to support the students if they needed clarification and checked the questionnaires for completeness on the spot.

### 2.4. Measurement

The outcome variables “psychoactive substance use” and “substance user’s level of dependency on the substance(s)” were measured using an adapted tool developed and validated by the WHO, the “Alcohol, Smoking and Substance Involvement Screening Test (ASSIST)” [24]. Status of Psychoactive substance use was determined from the first two questions in ASSIST tool: “In your life, which of the following substances have you ever used?” followed by (“In the past three months, how often have you used the substances you mentioned?”). Responses for these two questions were merged, which produces three mutually exclusive outcomes: (1) Never used in my lifetime, (2) previous user but not within the past three months and (3) current user (used in the past three months).

For current users, the last five questions of the ASSIST survey were merged and summed up to determine the level of dependency on psychoactive substance use. Each current substance user’s level of dependency on the psychoactive substance(s) was determined by summing the participant’s scores for the five items: (1) How often have you felt a strong desire or urge to use the substance(s) you use? (2) How often did your use of the substance(s) lead to health, legal, social or financial problems? (3) How often have you failed to do what was normally expected of you because of your substance use? (4) Has a friend or a relative or anyone else ever expressed concern about your use of the substance(s)? (5) Have you tried and failed to control, cut down or stopped using the substance(s)? The tool classifies the summed score for substances other than alcohol as “low risk of dependency” (score 0 to 3), moderate (scores 4 to 26) and high (scores of 27 and above). The range differs for a low and moderate risk of dependency on alcohol; low (0 to 10) and moderate (11 to 26) [24].

### 2.5. Data Analysis

Data were entered into EPi data version 3.2 then imported into SPSS version 21 software. Data were summarized using descriptive statistics. Multinomial logistic regressions were run to assess the factors associated with the three outcomes 1) never used in life time, here after referred as “Never user”; 2) previous user but not within the past three months, here after referred as “Ex-user”; and 3) used in the past three months, hereafter referred to as “Current user”. Factors that were found statistically significant with bivariate logistic regression at a *p*-value of 0.2 and contained at least 10 people per response item were included in a multivariable multinomial logistic regression model using backward stepwise selection. Variables were checked for multi-collinearity in the multivariate model and continuous predictors were checked for normality before further analysis. Model goodness-of-fit tests were completed for the final model (a *p*-value greater than 0.05). The highest proportion of missing values per variable was less than 2% and those participants were dropped out of the multivariate model (complete participant analysis). Adjusted odds ratios (AOR) were reported with 95% confidence intervals (95% CI). For current substance users, their level of dependency on the substance used was determined by summing the risk score gained from ASSIST.

### 2.6. Ethical Consent

Ethical approval was obtained from the ethical review board of the College of Health Science at the Research and Community Service office, Mekelle University (expedited approval number 1048/2017). Written consent was provided by each participant.

## 3. Results

### 3.1. Socio-Demographic Characteristics of Respondents

Of the 1220 students invited to participate, 1214 (99.5%) completed the questionnaire. Respondents’ mean age was 21 years (SD ± 1.8; range 18 to 29 years). Seventy four percent were male and 61.3% resided in an urban setting before joining the campus. Fifty five percent were from Tigray and 21% were from Amhara. Most (88.6%) were Orthodox Christians. Over half (54%) were from the Engineering Institute of Technology and 44% were first- and second-year students (Table 1).

### 3.2. Lifetime and Current Use of Psychoactive Substance

The lifetime prevalence of psychoactive substance use, which refers to those who have “ever used” a substance in their lifetime, was 66.5%, (95% CI = 64% to 69%), which was highest for alcohol (65%) and lowest for “other illicit drugs” (2%). Likewise, the lifetime prevalence for tobacco was 15%, khat 12% and cannabis 6%. The mean age of the respondents at first use of alcohol was 16 (standard deviation (SD) ± 3.9) years and 17 (SD ± 2.8) years for tobacco.

The current prevalence of psychoactive substance use was 49% (594; 95% CI = 46% to 51%). The highest current prevalence was 35.5% for alcohol followed by 7.8% for tobacco, 5.7% for khat and 2.7% for cannabis. Only 3.5% (95% CI = 2.6% to 4.7%) reported currently using more than one psychoactive substance. Of the participants, 18% (95% CI = 16% to 20%) were ex-users and 33% (95% CI = 31% to 36%) reported never having used psychoactive substances.

Daily use was reported by 15% (tobacco), 14% (cannabis) and 8% (khat). Weekly use was reported by 28% of those using alcohol, 25% using khat and 21% using cannabis.

### 3.3. Reasons for Psychoactive Substance Use

Of the 594 current users, 99.5% (*n* = 591) of them specified their primary reason for current psychoactive use while the remained three did not. Moreover, 86% reported relaxation as the primary reason for current use while 56% reported social and academic reasons (Table 2).

### 3.4. Dependency on Psychoactive Substance Use

Dependency, according to the WHO ASSIST screening scale, was indicated by 17% (95% CI = 14% to 20%) of current users (*n* = 591), who scored in the moderate to high risk range of dependence on substance(s). Specifically, 65 (11%) of current substance users were at moderate risk of dependency and 36 (6%) were at high risk of dependency.

### 3.5. Factors Associated with Psychoactive Substance Use

Being over 21 years of age, male, living in urban areas before university, Orthodox Christian, and in the 3rd year of study or above were significantly associated with current substance use. Students over 21 years of age were 1.6 times more likely to be a current user than those aged 21 and less (AOR = 1.6, 95% CI = 1.37 to 2.25). Males were more likely to be a current substance user than females who had never used (AOR = 3.16, 95% CI = 2.3 to 4.34). Students who come from an urban area were 2.4 times more likely to be a current user than those from a rural one (AOR = 2.39, 95% CI = 1.77 to 3.23). Orthodox Christian students were more likely to be a current user than non-orthodox Christian students (AOR = 7.55, 95% CI = 4.56 to 12.48). Students were more likely to be a current user in the 3rd (AOR = 2.3, 95% CI = 1.49 to 3.55), 4th (AOR = 2.0, 95% CI = 1.2 to 3.51) and 5th year (AOR = 4.0, 95% CI = 2.81 to 7.67) than in the 1st year of study (see Table 3).

In addition, males were 1.8 times more likely to have previously used substances than females (AOR = 1.89, 95% CI = 1.29 to 2.77) and Orthodox Christians 2.08 times more likely to have previously used than non-Orthodox Christians (AOR = 2.08, 95% CI = 1.26 to 3.41) (see Table 3).

## 4. Discussion

The lifetime prevalence of psychoactive substance use (alcohol, khat, tobacco, Hashish and illicit drugs) was 66.5% (95% CI = 64% to 69%) amongst Mekelle University students while the current prevalence was 49%. Of the students, 18% were previous users and 33% reported having never used psychoactive substances. Drinking alcohol was the most commonly reported behavior. Being male, Orthodox Christian, residing in an urban area before university, the more years enrolled at university and an older age were positively associated with current substance use.

A similarly high lifetime prevalence of substance use has been reported at other Universities in Ethiopia: 62.4% at Haramaya University [25], 45% at Sheba University College at Mekelle [26] and 64% at Dire Dawa University [14]. One study among college students in Western Kenya revealed a slightly higher (70%) lifetime prevalence of psychoactive substance use [27]. Similarly, other recent studies also reported a high prevalence of current substance use [12,15,19,28,29].

Despite students over 21 years of age being more likely to use substances, the average age of first trying alcohol and tobacco was less than 18 years. A previous study from India had similar findings [30]. Though consistent use and its negative consequences often begin during adulthood, adolescence is a key period of experimentation [31]. Programs that are designed to target and reduce adolescents’ drive to experiment with substances [31] and incorporate brief interventions before substance use becomes problematic, may reduce substance use [32]. We also found current use increased with seniority of students. A study in Hawassa University found seniority was positively associated with khat use with the highest current use among forth year students and above [29]. Similar findings were reported in Haramaya University of Ethiopia [26], Sudan [33] and Nigeria [19]. Sustained peer pressure, familiarity with how to access substances and the varieties available, as well as poor academic performance may explain the increase in substance use over time [21,32,33].

Of particular concern was that 17% of current psychoactive substance users were found to be at risk of dependency. Similarly, two in ten students in Bahir Dar University practiced harmful khat use [17] and one in ten students in Spanish universities were substance dependent [34]. A recent study from Wolayta Sodo University of Ethiopia also reported that 11.4% of the sampled students were problematic alcohol users [33]. An American study found 12.7% of their college student population were heavy drinkers [35]. Problematic use of psychoactive substances has been correlated with students’ poor academic performance, unprotected sex and hospital admission attributed to substance-induced health problems [36]. The WHO also suggests that individuals at risk of being dependent on psychoactive substances should be offered intensive behavioral and medical interventions [24].

Our finding that males are more likely to report current psychoactive substance use has been found before [15,16,19,20,25,26,27,28]. The WHO (2011) reports a fivefold greater risk of death attributable to substance use [37] and a twofold greater risk of substance use disorder [38] among men than women. For a variety of social and cultural reasons, substance use may be considered more permissible for males than females in many societies. Twenty five percent of our students mentioned peer pressure as reason to use psychoactive substances, which is a frequently cited determinant of psychoactive substance use [33,39]. Previous studies have shown that gender-specific interventions to reduce substance use are more effective [40].

Studies in Bahir Dar and the Sheba University of the Mekelle Branch have also found higher rates of substance use among students of urban and semi-urban origin [20,26]. Urbanity has been positively associated with the perception of ease of access and the use, as well as females’ harmful use of psychoactive substances [39,41,42,43]. In contrast, rural students in Canada were more likely to consume alcohol [44], and methamphetamine in India [45], than urban students. Thus, an urban vs. rural classification may be a proxy measure of the substances that are easy to access and use rather than a strong indicator of use. Those not able to access psychoactive substances will never use them nor become dependent upon them.

Being Orthodox Christian was also positively associated with psychoactive substance use. Most (88.6%) of our participants were Orthodox Christians. They had the highest lifetime (65%) and current (35.5%) prevalence for use, which was highest for alcohol. This finding is consistent with results of previous studies on alcohol use [25,46]. In contrast, studies on khat use indicated that being Muslim was associated with khat use [15,20,28]. Other studies also reported religious reasons for some of psychoactive substance use [25]. Despite religiosity being a protective factor for early substance use, religious sanctions and permissibility of substance use varies markedly across belief systems.

A qualitative part of the same project also shows that feeling helpless following detachment from family, prior experience with substances, socialization reasons, low academic performance, physical environment (explained by easy access to substance and limited recreational alternatives) and sub-optimal organizational support [22]. The high prevalence of psychoactive substance use, an array of the factors from the qualitative and quantitate factors and the high dependency rate on the substance implies the need to avail a continuum of early prevention strategies from high school to universities.

### Limitations of the Study

Our findings rely on students’ self-report, which may be less precise than objective measurements given cultural sensitivity and social acceptance around substance use in Ethiopia. As a higher proportion of participants were males (73.7%) and Orthodox Christian (88.3%), these non-equivalent comparative groups (sex and religion) should be taken into account. In addition, substance use may sometimes be condoned the user’s religious affiliation. Therefore, further studies that take such factors into account might supplement the current findings. Studies that assessed more extensive risk factors using students from more universities could complement the findings in the current study.

## 5. Conclusions

Approximately half of the studied Mekelle University undergraduate students use psychoactive substances, with 17% at risk of dependency. The likelihood of current substance use increased with seniority, being male, originally residing in urban areas and being over 21 years of age. Evidence-based prevention strategies are needed for school-aged children to prevent psychoactive substance first use, and at university to prevent dependence on substances. Interventions designed to stop current psychoactive substance use may also have promise for reducing dependency.

## Figures and Tables

**Table 1 ijerph-17-00847-t001:** Socio-demographic characteristics of Mekelle University undergraduate students in Ethiopia in 2017 (*n* = 1214).

Variables	Categories	*n*	(%)
Age	>21 years	517	42.6
	≤21 years	682	56.8
Sex	Male	889	73.7
	Female	318	26.3
Residence	Urban	744	61.3
	Rural	469	38.7
Ethnicity	Tigray	673	55.4
	Amhara	257	21.3
	SNNP	55	5.0
	Oromo	49	4.0
	Others *	57	4.7
Religion	Orthodox Christian	1073	88.3
	Muslim	55	4.5
	Protestant	65	5.4
	Others **	18	0.15
Year of study	5th year	166	13.7
	4th year	204	16.8
	3rd year	306	25.2
	2nd year	258	21.3
	1st year	280	23.1
Currently used substances	Alcohol	431	35.5
	Tobacco	95	7.8
	Khat	69	5.7
	Illicit drugs	32	2.7
Use status	Current prevalence	594	48.9
	Past (ex-users)	218	17.9
	Never users	402	33.1
Period prevalence	Life time/ever use of any substance	807	66.5

* Afar, Somali ** Catholic.

**Table 2 ijerph-17-00847-t002:** Reasons for current psychoactive substance use among Mekelle University undergraduate students in Ethiopia in 2017.

Reasons for Psychoactive Use	Percentage (*n* = 591)
Entertainment/relaxation	86
Social and academic reasons	56
Friends use	25
Family uses	11
Accessible	6
To stay awake	9

**Table 3 ijerph-17-00847-t003:** Multiple multinomial logistic regression on factors associated with psychoactive substance use among Mekelle University undergraduate students in Ethiopia in 2017.

Variables	Univariable		Multivariable	
	Current User *n* = 591 COR (95% CI)	Ex-User *n* = 218 COR (95% CI)	Current User *n* = 591 AOR (95% CI)	Ex-User *n* = 218 AOR (95% CI)
**Age**				
Above 21 years of age	1.9 (1.46, 2.47) **	1.13 (0.80, 1.59)	1.6 (1.37, 2.25) **	1.05 (0.64, 1.71)
**Sex**				
Male	1.81 (1.25, 2.63)	2.46 (1.85, 3.28)	3.16 (2.30, 4.34)	1.89 (1.29, 2.77) *
**Residence**				
Urban	1.7 (1.31, 2.22) **	0.92 (0.66, 1.29)	2.39 (1.77, 3.23) **	1.08 (0.76, 1.54)
**Religion**				
Orthodox	5.97 (3.74, 9.52) **	1.77 (1.11, 2.82)	7.55 (4.56, 12.48) **	2.08 (1.26, 3.41) *
**Year of Study**				
≥5th year	4.27 (2.67, 6.84) **	0.93 (0.48, 1.80)	4.0 (2.18, 7.67) **	0.93 (0.41, 2.07)
4th year	2.26 (1.49, 3.43) **	1.07 (0.64, 1.80)	2.0 (1.2, 3.51) **	0.99 (0.52, 1.90)
3rd year	2.09 (1.44, 3.03) **	0.96 (0.61, 1.51)	2.3 (1.49, 3.55) **	0.97 (0.58, 1.62)
2nd year	1.25 (0.85, 1.83)	0.86 (0.54, 1.36)	1.09 (0.77, 3.23)	0.81 (0.51, 1.54)

Key: COR = Crude Odds Ratio, AOR: Adjusted Odds Ratio. Reference category: for the dependent variable is “Never” and for the independent variables ≤21 years; female; rural; other religions (combined); 1st year of study; * *p*-value < 0.05; ** *p*-value < 0.01.

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
