# Peer review of "Prevalence of, Factors Associated with and Level of Dependence of Psychoactive Substance Use among Mekelle University Students, Ethiopia"

_ijerph, 2020, doi:10.3390/ijerph17030847_

Round 1

Reviewer 1 Report

There are a lot of grammatical mistakes in the text, lots of inaccuracies, e.g. in line 72 the source is DOI and not the second name, publishing year, although it's a manuscript, references are inaccurate - no author of manuscript, no first name, no journals have been mentioned or pages, etc. The WHO source is listed but no such information is available in the specific source. Sources are mixed - 23 source has been listed as Mekelle University, but in the references it is the same as WHO. Therefore all the other sources are already obliterated. The same source has been mentioned twice (34 and 47).

The references should be organized into a single system, currently they are chaotic!

The introduction part has unnecessary information on consequences and mortality that does not match the topic of manuscript.

It has not been explained why the authors highlight Khat.

With regard to the selection of respondents, it is not clear what "Randomization was done by students from their respective departments" means. How was it done? It is also unclear how the sample was taken when 55% of the 4 colleges were obtained from the Engineering Institute of Technology. The sampling and the resulting sample should be specified. The section "Sample size and sampling procedure" does not describe the research sample that was then studied, this information is presented as results, but it is not results.

There is also some uncertainty about the research instrument (section 'Data collection process and instrument'). If it is an author's instrument, as it seems to imply from the text, then it is not clear why it should be translated into English and back into Amharic (the Ethiopian national official language). It is not described how many questions the questionnaire contained, how long it took to complete, whether everyone filled in or refused, if so how many? etc. The authors further state that ASSIST contains five questions, but the originals of this instrument have eight questions, so it is not clear which questions did authors use.

Inaccuracies in the data presented. How do you get the Life time / ever use of any substance - 807 respondents? Table 1 - current prevalence has 594 respondents, but Table 2 - 591! There are 1218 Users (current, past, never), but the authors indicate that only 1214 (99.5%) completed the questionnaire. It is also not clear why the authors break down the age to 21 and after 21, what is the explanation?

Author Response

Reviewer 1

Comments and Suggestions for Authors

Comments: There are a lot of grammatical mistakes in the text, lots of inaccuracies, e.g. in line 72 the source is DOI and not the second name, publishing year, although it's a manuscript, references are inaccurate - no author of manuscript, no first name, no journals have been mentioned or pages, etc. The WHO source is listed but no such information is available in the specific source. Sources are mixed - 23 source has been listed as Mekelle University, but in the references it is the same as WHO. Therefore all the other sources are already obliterated. The same source has been mentioned twice (34 and 47).

The references should be organized into a single system, currently they are chaotic!

Authors’ response: Thank you for the suggestions. We have look at each reference for appropriate citation.  

Comments: The introduction part has unnecessary information on consequences and mortality that does not match the topic of manuscript.

Authors’ response: We have limited to “negative social, physical, and mental health consequences”.

Comments: It has not been explained why the authors highlight Khat.

Authors’ response:

Comments: With regard to the selection of respondents, it is not clear what "Randomization was done by students from their respective departments" means. How was it done? It is also unclear how the sample was taken when 55% of the 4 colleges were obtained from the Engineering Institute of Technology. The sampling and the resulting sample should be specified. The section "Sample size and sampling procedure" does not describe the research sample that was then studied, this information is presented as results, but it is not results.

Authors’ response:  The sample was 1220. This was proportionally allocated to 15 randomly selected departments. After allocation, we used computer generated randomization to select the allocated number of students from each department. Definitely, the higher proportion of students from Engineering is the reflection of higher number of students per department compare to others. There is high number of engineering students in the campus similar to most of Ethiopian Universities including Mekelle University nowadays. To make this procedure clear, we have reconstructed the paragraph as;      

The list of registered students in the selected departments was obtained from University’s registrar office to prepare sample frame for each department. Consequently, the required samples were proportionally allocated to the number of the students in each selected department based on the list provided from the registrar office. Finally, computer generated random sampling was employed to select study participants from respective departments to ensure random selection.”

Comments: There is also some uncertainty about the research instrument (section 'Data collection process and instrument'). If it is an author's instrument, as it seems to imply from the text, then it is not clear why it should be translated into English and back into Amharic (the Ethiopian national official language). It is not described how many questions the questionnaire contained, how long it took to complete, whether everyone filled in or refused, if so how many? etc. The authors further state that ASSIST contains five questions, but the originals of this instrument have eight questions, so it is not clear which questions did authors use.

Authors’ response: The tool consists of three sub-sections. (1) socio-demographics (2) self-reported substance use (3) level of dependency. We have adopted five Questions from ASSIST to measure the level of dependency, other questions from other literature and expert opinions. Now, if the previous paragraph was not clear to provide the intended message, we have rephrased it.      

Comments:  Inaccuracies in the data presented. How do you get the Life time / ever use of any substance - 807 respondents? Table 1 - current prevalence has 594 respondents, but Table 2 - 591! There are 1218 Users (current, past, never), but the authors indicate that only 1214 (99.5%) completed the questionnaire.

 Authors’ response: To make the data clear. The ample size calculated was 1220. Of these approached, 1214 completed the questionnaire, which yield response rate of 99.5%. Of the 1214 study participants, 807(66.5%) reported life time substance use, 594(48.95%; rounded to 49%) reported current use, 218(17.95%; rounded to 18%) and 402(33.1%: rounded to 33%) reported have never used. Of the 594 current user, 591 of them specified their reason for the use while 3 didn’t (as missed variable). That is why we calculated the percentage out of 591. We made clear in the text [line 193-195] to avoid confusion. Misleading seems introduced as due to typological error in table 1, the never users (402 not 406) in the previous version, is corrected now. We hoped the confession is now resolved.   

Comments: It is also not clear why the authors break down the age to 21 and after 21, what is the explanation?

 Authors’ response: Literature shows that psychoactive substance misuse is initiated at early age. As we are considering University students, this age category will give clue if students have initiated before or after joining University. However, most intervention to help quitting substance use focuses on older ages. Besides, the percentage distribution of participant’s age, the mean was 21 years.  If this justification doesn’t seem good, we will be looking for your feedback.

Reviewer 2 Report

The article submitted for review: “Prevalence of, factors associated with and level of Dependence of Psychoactive Substance Use among Mekelle University Students, Ethiopia” is a typical scientific analysis regarding the presentation of the scale of prevalence, factors and the level of addiction of psychoactive substances among students.
The article was correctly formatted, divided into parts in accordance with the requirements of the journal. It has been correctly written and has properly selected literature. The authors correctly presented the research topic, which has been presented to the reader in an accessible form.
It is worth supporting the diligence of the publication before publication - which I recommend to authors for supplementing the article with following issues:
- in-depth analysis of the definition of addiction in the context of entries in DSM-V and ICD-11
- directions of prevention of addiction to adults
- clarifying in the limitations that the own report from this study is based on non-equivalent comparative groups regarding sex (73.7% men) and religious groups (88.6% Orthodox Christians) which may have (and as the content of the article shows) influence on drawing correct conclusions from research
- reformulation or deletion of records related to test results that are associated with drawing conclusions based on data from a comparison of non-equivalent groups. This is a methodological error, the dimension of which has social important and sensitive consequences (verse: 241-216; 258; 275). I suggest that the authors re-calculate the data and not include religion in the regression model.
The article can be published, however, after careful correction by the authors.

Author Response

Reviewer 2

Comments: in-depth analysis of the definition of addiction in the context of entries in DSM-V and ICD-11

Authors’ response:
Comments:- directions of prevention of addiction to adults

Authors’ response:
Comments:- clarifying in the limitations that the own report from this study is based on non-equivalent comparative groups regarding sex (73.7% men) and religious groups (88.6% Orthodox Christians) which may have (and as the content of the article shows) influence on drawing correct conclusions from research

Authors’ response: Thank you for the suggestion this beautifully said. We accepted this suggestion and we included in the limitation.
Comments:- reformulation or deletion of records related to test results that are associated with drawing conclusions based on data from a comparison of non-equivalent groups. This is a methodological error, the dimension of which has social important and sensitive consequences (verse: 241-216; 258; 275). I suggest that the authors re-calculate the data and not include religion in the regression model. 

Authors’ response: We have carefully looked at the data related to the religion. It is clear that the proportions are non-equivalent. We have included this in the limitation of the study. However, we opt to let this as it is also reported y literature.  

Reviewer 3 Report

The work entitled “Prevalence of factors associated with and level of Dependence of Psychoactive Substance Use among Mekelle University Students, Ethiopia” addresses a topic of interest, since it is a public health problem in a population that should theoretically be more protected from the use of psychoactive substances, the university population.

The work has quality, but there are a series of limitations that should make me concern about its viability to be published as it is currently.

Major issues

Title:

This article includes in the title the term “prevalence”. This prevalence is obtained by means of a few self-reported questions. I consider a little bit pretentious to talk about a prevalence study when only self-report questions are used and also in a sample from a single university in a city from Ethiopia. Perhaps it would be more correct to talk about frequency or at least point out much more explicitly in the subsection “Limitations of the study”.

Introduction/background

Regarding the rationale of the work, this work is well justified, with an updated review of related works that allow the discussion of the results. However, I believe that better justification should be given as to why the frequency/prevalence of use and dependence should be known and what this work provides that is completely new and different to previous studies.

Method

There are several problems in this section.

The first is that, although the classification of use and dependence of psychoactive substances was made with the ASSIST instrument, it should be noted whether such classification followed ICD-10 diagnostic criteria.

The second problem is that it is not possible to understand why use and dependence is reported and not abuse. It should be indicated whether it is because of a limitation of the instrument or because the authors decided not to report this data and why.

The third aspect is that the sample is recruited exclusively from a city university. This is a limitation that should also be highlighted in the “Limitations of the study” section, that is surprisingly brief.

The fourth aspect is that the instruments subsection does not describe how the factors associated with substance use and dependence were assessed or measured.

The fifth point, and in the same line of the previous one, the limitations should also indicate that the number and variety of risk factors analyzed is insufficient and limited.

The Results and Discussion sections are well written, and I have no objections.

References

The references should be checked. There are many errors, since the style guidelines of the journal were not followed. This is the section with the most errors of style in the whole work. The correspondence between quotations and references is not given, due to possible misprints.

Let me point out only four examples:

Reference numbers 7 and 8 seem to be the same, separated by error. Reference number 22 must be wrong. Reference 23 is not right. Reference 44 is incomplete.

Minor issues

Page 1, line 19. There is an extra space between "Ethiopia." and "We aimed”.

Page 1, line 28. The inclusion of N=1214, used to calculate prevalence, is confusing, because it appears that the number of people with drug use was 1214 and actually refers to the total N.

Page 2, line 71-72. The authors include the following sentence "A qualitative exploration study of the same project is "(available at: https://doi.org/10.1186/s13011-018-0190-1). The inclusion of the link to doi of an article to indicate that it is available does not seem appropriate or necessary.

Page 2, line 84. A comma is missing in front of "(2) identify factors"

Page 3, line 107. There seems to be a space between "use." and "The questions asked..."

Line 114. There is an extra space between "in English." and "One language."

Page 4, line 142. There is an extra space between "descriptive statistics." and "Multinomial logistic..."

Page 7, line 275. I should be included a dot after "use in Ethiopia".

Author Response

Reviewer 3

Comments and Suggestions for Authors

The work entitled “Prevalence of factors associated with and level of Dependence of Psychoactive Substance Use among Mekelle University Students, Ethiopia” addresses a topic of interest, since it is a public health problem in a population that should theoretically be more protected from the use of psychoactive substances, the university population.

The work has quality, but there are a series of limitations that should make me concern about its viability to be published as it is currently.

Major issues

Title:

Comments: This article includes in the title the term “prevalence”. This prevalence is obtained by means of a few self-reported questions. I consider a little bit pretentious to talk about a prevalence study when only self-report questions are used and also in a sample from a single university in a city from Ethiopia. Perhaps it would be more correct to talk about frequency or at least point out much more explicitly in the subsection “Limitations of the study”.

 Authors’ response: Thank you for the suggestion. With the limitation we apparently discussed in the limitation section, we would argue that prevalence can still goes with the intention of the study and the measurement used. However, we have included in these limitations could be better addressed by further studies that consist of that extensive risk factors and students from University. The paragraph is paraphrased as, 

“Our findings rely on students’ self-report, which may be less precise than objective measurement given cultural sensitivity and social acceptance around substance use in Ethiopia. As higher proportion of participants were males (73.7%) and Orthodox Christian (88.3%), this non-equivalent comparative groups (sex and religion) should be taken into account regarding the conclusion drawn. In addition, substance use may sometimes be condoned the user’s religious affiliation. Therefore, further studies that take such factors into account might supplement the current findings. Studies that assessed more extensive risk-factors using students from more number of Universities could complement the determined prevalence reveled in the current study.”

Comments:

Introduction/background: Regarding the rationale of the work, this work is well justified, with an updated review of related works that allow the discussion of the results. However, I believe that better justification should be given as to why the frequency/prevalence of use and dependence should be known and what this work provides that is completely new and different to previous studies.

Comment:

Method There are several problems in this section. The first is that, although the classification of use and dependence of psychoactive substances was made with the ASSIST instrument, it should be noted whether such classification followed ICD-10 diagnostic criteria.

 Authors’ response: The ASSIST is the Alcohol, Smoking and Substance Involvement Screening Test, which is a brief screening questionnaire to find out about people’s use of psychoactive substances. It is a manual introduced by WHO in 2005.  The ASSIST can be used in a number of ways to assess patients’ substance use which is particularly important for primary care settings where a high proportion of patients are likely to be substance users, e.g., university health services (The ASSIST Manual, 2005).  That is why we have used the manual. The instrument provides directions regarding calculating a Specific Substance Involvement Score for each substances which goes in line with the ICD-10.

Comment: The second problem is that it is not possible to understand why use and dependence is reported and not abuse. It should be indicated whether it is because of a limitation of the instrument or because the authors decided not to report this data and why.

 Authors’ response: This measurement was enabled us to determine life time and current prevalence of psychoactive substance use as well as the level of dependence to the substances under use.

Comment: The third aspect is that the sample is recruited exclusively from a city university. This is a limitation that should also be highlighted in the “Limitations of the study” section,that is surprisingly brief.

 Authors’ response: Thank you for rising the concern. We have included this information in the revised version.

Comment: The fourth aspect is that the instruments subsection does not describe how the factors associated with substance use and dependence were assessed or measured.

Authors’ response: To make this clear for the readers, we re-wrote the paragraph that states how the outcome variables (use) and dependence were measured. Psychoactive substance use had three outcomes (Current user, ex-user and never users). Multivariable multinomial logistic regression model was applied to identify factors associated with psychoactive substance use status. This s detailed in the data analysis section.    

Comment: The fifth point, and in the same line of the previous one, the limitations should also indicate that the number and variety of risk factors analyzed is insufficient and limited.

Authors’ response: This in indicated in the current version

References

The references should be checked. There are many errors, since the style guidelines of the journal were not followed. This is the section with the most errors of style in the whole work. The correspondence between quotations and references is not given, due to possible misprints.

Let me point out only four examples:

Reference numbers 7 and 8 seem to be the same, separated by error. Reference number 22 must be wrong. Reference 23 is not right. Reference 44 is incomplete.

Minor issues 

Page 1, line 19. There is an extra space between "Ethiopia." and "We aimed”.

Page 1, line 28. The inclusion of N=1214, used to calculate prevalence, is confusing, because it appears that the number of people with drug use was 1214 and actually refers to the total N.

Page 2, line 71-72. The authors include the following sentence "A qualitative exploration study of the same project is "(available at: https://doi.org/10.1186/s13011-018-0190-1). The inclusion of the link to doi of an article to indicate that it is available does not seem appropriate or necessary.

Page 2, line 84. A comma is missing in front of "(2) identify factors"

Page 3, line 107. There seems to be a space between "use." and "The questions asked..."

Line 114. There is an extra space between "in English." and "One language."

Page 4, line 142. There is an extra space between "descriptive statistics." and "Multinomial logistic..."

Page 7, line 275. I should be included a dot after "use in Ethiopia".

Author’s response: All of these comments are addressed in the current version.